# Single-Cell Technologies to Study Phenotypic Heterogeneity and Bacterial Persisters

**DOI:** 10.3390/microorganisms9112277

**Published:** 2021-11-01

**Authors:** Patricia J. Hare, Travis J. LaGree, Brandon A. Byrd, Angela M. DeMarco, Wendy W. K. Mok

**Affiliations:** 1Department of Molecular Biology & Biophysics, UConn Health, Farmington, CT 06032, USA; hare@uchc.edu (P.J.H.); lagree@uchc.edu (T.J.L.); bbyrd@uchc.edu (B.A.B.); ademarco@uchc.edu (A.M.D.); 2School of Dental Medicine, University of Connecticut, Farmington, CT 06032, USA; 3School of Medicine, University of Connecticut, Farmington, CT 06032, USA

**Keywords:** antibiotic persistence, phenotypic heterogeneity, single-cell analysis

## Abstract

Antibiotic persistence is a phenomenon in which rare cells of a clonal bacterial population can survive antibiotic doses that kill their kin, even though the entire population is genetically susceptible. With antibiotic treatment failure on the rise, there is growing interest in understanding the molecular mechanisms underlying bacterial phenotypic heterogeneity and antibiotic persistence. However, elucidating these rare cell states can be technically challenging. The advent of single-cell techniques has enabled us to observe and quantitatively investigate individual cells in complex, phenotypically heterogeneous populations. In this review, we will discuss current technologies for studying persister phenotypes, including fluorescent tags and biosensors used to elucidate cellular processes; advances in flow cytometry, mass spectrometry, Raman spectroscopy, and microfluidics that contribute high-throughput and high-content information; and next-generation sequencing for powerful insights into genetic and transcriptomic programs. We will further discuss existing knowledge gaps, cutting-edge technologies that can address them, and how advances in single-cell microbiology can potentially improve infectious disease treatment outcomes.

## 1. Introduction

In a world of diverse threats, phenotypic heterogeneity is a bet-hedging strategy that increases the odds of survival for clonal bacterial populations. In a given cohort, some bacteria may be slow-growing and better prepared to survive external threats, while others are metabolically poised to take advantage of nutrient windfalls for more rapid propagation. Because of this, external insults such as antibiotic treatment may not fully eradicate a bacterial population even if it appears genetically susceptible [1]. Some bacteria can endure antibiotic treatment until the insult ends and they can resume growth and repopulate.

These resurgent cells, termed persisters, contribute to recurrent or recalcitrant infections that are difficult to resolve [2,3]. Clinically, persisters are implicated in the chronicity of a range of infections, including urinary tract infections by uropathogenic *Escherichia coli*, pneumonia in cystic fibrosis patients from *Pseudomonas aeruginosa*, or tuberculosis from the namesake pathogen, *Mycobacterium tuberculosis* [4]. Antibiotic resistance is a widely recognized threat to public health, but with increasing evidence suggesting that persistence begets resistance, it is clear that persisters present a multifaceted challenge in clinical infection management [5,6,7,8].

Better understanding of bacterial persistence can facilitate the development of anti-persister treatment strategies; however, identifying and studying persistent bacteria is a complex endeavor. Unlike resistant bacteria, persisters lack distinct genetic identifiers and therefore appear indistinguishable from their susceptible kin [2]. Furthermore, population-based methods may mask the persistent subpopulations that are estimated to comprise only 0.001–1% of a population [9]. Furthermore, phenotypic states are inherently transient and shift in response to environmental conditions; therefore, it is even more important that chosen techniques faithfully capture physiological states with minimal cellular perturbations [10]. The field is currently limited by an inability to predict which cells will die, which will survive treatment and reawaken, or which cells will remain viable through treatment but will fail to resuscitate and replicate after treatment cessation [11]. The use of tools to track phenotypic heterogeneity, together with functional assays, can help elucidate which cells persist and how they do so.

Here, we provide an overview of single-cell techniques that are applicable to studying phenotypic heterogeneity and bacterial persisters (Figure 1). To avoid redundancy with a similar review, we will highlight cutting-edge technologies from the past five years and discuss improvements in classical techniques that can illuminate the aspects of persister biology that elude population-based methods [12].

## 2. Illuminating Single-Cell Traits with Fluorescence

### 2.1. Fluorescent Biosensors

Fluorescence is the foundation of many biological assays and can be measured in a variety of ways, including flow cytometry, microscopy, or spectrophotometry. One effective way to study phenotypic heterogeneity is with fluorescent reporter plasmids to indicate relative transcription levels of specific genes (Figure 2A) [12,13,14]. Many studies have utilized fluorescent reporters to study heterogeneity within vital bacterial genes related to persistence, growth, and other phenotypic characteristics [15,16,17,18,19]. Others have developed whole-genome reporter libraries to serve as a resource to better resolve transcriptional distinctions between cells [13]. Additionally, cutting edge “multi-reporter” constructs have been effectively optimized in *E. coli* and *P. aeruginosa* to enable observation of multiple transcripts simultaneously while subverting traditional limitations of spectral overlap (Figure 2A) [20,21,22]. 

Gene expression reporter plasmids are a reliable tool in analyzing transcription patterns, but additional strategies are needed to measure subsequent translation. Proteins are commonly identified by fluorescently tagged antibodies that bind specific protein epitopes, also allowing for protein abundance and localization studies [23]. Studying a protein of interest can also be accomplished with a transcriptional fluorescent protein (FP) fusion modifying a chromosomal or plasmid-borne copy of a gene (Figure 2B) [24,25,26]. This process has been scaled to the whole-genome level with the development of fluorescent fusion libraries in *E. coli*, among other species, that do not have an observable impact on cell growth [27,28].

Förster resonance energy transfer (FRET) is a protein-FP fusion technique that utilizes a pair of photoexcitable probes to indicate when specific proteins or molecules interact, allowing for the gathering of information on signaling pathways and protein localization [29,30]. Mechanistically, FRET involves the energy transfer from one probe that will excite its proximal partner, resulting in emission at a distinct wavelength (Figure 2B). If the tagged proteins do not interact, the pair of probes will be too far apart and the second probe will not receive the proper excitation wavelength to fluoresce. Keegstra et al. highlighted the value of FRET in studying phenotypic heterogeneity when they utilized FP-bound CheY and CheZ to demonstrate that chemotaxis signaling in *E. coli* is heterogeneous [30].

Protein-FP fusions continue to remain a reliable, efficient, and easily scalable strategy for studying proteins; however, this approach does come with its limitations. The addition of the FP can potentially lead to undesired protein folding, aggregation, or localization. Additionally, this technique is only feasible in genetically tractable species, precluding its use with most environmental or clinical isolates [25].

Small molecule fluorescent probes can be applied to a variety of bacterial systems to resolve single-cell morphological, metabolic, or signaling information (Figure 2C). Probes that bind nucleic acids (e.g., 4′,6-diamidino-2-phenylindole (DAPI), SYTOX Green, and Hoechst 33342) can indicate DNA content and are broadly used [23,31,32,33]. For example, in a study by Murawski and Brynildsen, the authors stained DNA with Hoechst 33342 and demonstrated that higher genome copy number correlates with increased persistence to the fluoroquinolone antibiotic, Levofloxacin, but that monoploid cells can still survive [34].

Oligonucleotide probes are commonly used to recognize specific RNA transcripts via fluorescent in situ hybridization (FISH) (Figure 2D). FISH protocols have been developed in recent years to bind specific RNA molecules in living, non-fixed bacteria, meaning this protocol could be adapted to studying gene expression heterogeneity in live cells over time [35]. Furthermore, combinatorial FISH using multiple RNA probes in parallel dramatically increases the transcriptomic capabilities of this classic technique. One variation of this approach, called par-seqFISH, was recently implemented to study heterogeneous gene expression of single cells based on their geographic location within a microbial population [36].

Existing biomolecules can also be leveraged in the development of new probes. Fluorescently labelled amino acids incorporated into newly synthesized peptidoglycan can indicate cell wall biosynthesis rates (Figure 2C) [37,38]. Similarly, antibiotic analogs that bind native targets have been engineered into biosensors. For instance, the puromycin analog O-propargyl-puromycin (OPP) was cleverly modified via click chemistry to be able to bind a fluorophore; OPP biosensor incorporation into nascent peptides could therefore be measured to convey single cell translation rates (Figure 2C) [39,40]. However, OPP biosensors are incompatible with intrinsically puromycin-resistant Gram-negative species, thus highlighting the need to consider whether biosensors will reliably colocalize with structures of interest for a given model.

Modern biologists have taken advantage of naturally occurring binding motifs to create ligand-specific biosensors. Notable contributions have been made towards the quantification of intracellular ATP, an essential metabolite at the crux of definitions of cell dormancy, viability, and persistence [41,42,43]. To study physiologically relevant concentrations of intracellular ATP at single-cell resolution, Yaginuma et al. optimized the ATP synthase epsilon subunit from *Bacillus* PS3 into the “QUEEN” ATP biosensor (Figure 2B). Intracellular ATP concentrations are measured by exciting the sample at two distinct wavelengths and calculating the ratio of the emission intensities [44]. To facilitate imaging analysis using a single excitation wavelength, Lobas et al. developed the ATP sensor iATPSnFR and demonstrated its compatibility with confocal microscopy for imaging and quantifying ATP within single cells [45]. Note, however, that this sensor was used in mammalian cells and so certain features, such as the integration of the sensor into the plasma membrane via a specific trafficking vector, will require further optimization to be amenable in bacteria.

In addition to proteins, functional nucleic acids have also been engineered into fluorescent biosensors. For example, riboswitch-based biosensors have been developed for specific single-cell analyses. Riboswitches are regulatory RNAs found in both eukaryotes and prokaryotes and are remnants of ancestral RNA-centric organisms [46,47]. These RNA elements can be found in untranslated regions of messenger RNAs and they consist of a ligand binding aptamer domain along with a regulatory expression platform. In response to binding of a specific ligand in the aptamer domain, these small probes change conformation and can act like the switch on a DNA railroad track, diverting RNA polymerase from its default transcriptional pathway to turn transcription either “on” or “off”. Riboswitch conformational changes can also regulate translation by blocking ribosomal binding. Taking advantage of these regulatory elements, biologists have engineered riboswitches for use as intracellular biosensors (Figure 2D). Kellenberger et al. leveraged riboswitch biology to engineer a probe for detecting intracellular cyclic di-GMP, a signaling molecule with critical roles in regulating virulence, planktonic versus biofilm lifestyles, and antibiotic persistence [48,49]. Studying this key intracellular molecule was made possible by combining a c-di-GMP-recognizing riboswitch aptamer to Spinach, another aptamer with a chromophore binding pocket. In this fusion biosensor, binding of c-di-GMP causes a conformational change such that Spinach’s binding pocket becomes accessible for binding and activating the chromophore 3,5-difluoro-4-hydroxybenzylidene imidazolinone (DFHBI). The fluorescent signal can then be analyzed to determine secondary messenger activity in live cells. This combinatorial approach can be applied for creating biosensors from other naturally occurring aptamers or, in theory, recognition aptamers could be rationally designed for highly specific ligand detection.

The application of riboswitch biosensors has taken off in the last decade. They have been used in both Gram-negative and Gram-positive bacteria; have been implemented in the study of intracellular secondary messengers, amino acids, and nucleobases; and have been designed using a variety of natural and synthetic aptamers (interested readers should refer to a recent review by Husser et al. for details) [50]. Given the importance of intracellular metabolite concentrations in persister formation, this technology could vastly expand our understanding of how specific small molecules and metabolites contribute to persister physiology and heterogeneity [42,43].

### 2.2. Flow Cytometry

In the pursuit of highlighting distinctions in heterogeneous bacterial populations, flow cytometry remains a reliable and versatile technique in the analysis of single cells. First, the fluidic system injects the cells and buffered solutions into flow lines, with differential pressure allowing the cells to be focused into a single-file line [51,52]. Cells are then directed into the path of an excitation device, which can be used to measure size, granularity (the material inside the cell), and various fluorescent properties of the cell. The photodetectors in cytometers are able to detect photons emanating from individual cells, allowing for the characterization of these properties at a single-cell level [51,52]. In recent decades, flow cytometry has been utilized to provide insights into optical and fluorescence-based cellular properties of single eukaryotic and prokaryotic cells.

One of the most influential developments in flow cytometry was the deployment of fluorescent-activated cell sorting (FACS). When flow cytometry is interfaced with FACS, cells showing a desired fluorescent characteristic can be differentially charged and isolated through the use of electrical currents and electromagnetic devices [53,54]. This technique is crucial in the study of bacterial phenotypic heterogeneity because of its ability to resolve and isolate single cells for further analysis. These appropriately sorted cells can serve multiple purposes, including the return to healthy growing conditions for further division, or for direct analysis through microscopy and other techniques [55]. For example, researchers have used mCherry and Redox Sensor Green (RSG) to sort cells based on growth and metabolic activity, respectively [56,57]. Similar studies used FACS to sort cells based on reporters for persistence-implicated genes following treatment with antibiotics, and then utilized those cells for additional biological assays and sequencing [58,59]. Other experiments have utilized FACS to study biomarkers implicated in persistence and other phenotypes of bacterial heterogeneity [43,60]. For example, previous studies have classified cells by growth or metabolic rates (through the use of fluorescent reporters of gene expression levels) and then tested the cell’s ability to endure various stressors [43,56,57,60].

These fluorescence-based techniques are remarkably effective in the analysis of bacterial identity, development, and physiology, with the capability of differentiating heterogeneous populations. Nonetheless, utilizing these fluorescence-based techniques with flow cytometry or FACS lacks an important level of informational resolution, such as the localization of proteins and other molecules, and the timing of fluorescently tracked physiological events. Because of this, traditional flow cytometry has been interfaced with additional instrumental methods to delve deeper into heterogeneity at the single-cell level.

In order to impart additional resolution in distinguishing individual cells among heterogeneous populations, fluorescent microscopy has been interfaced with flow cytometry in a technique referred to as imaging flow cytometry (IFC). IFC captures multiple images of cells as they move through the flow line [61]. IFC differs from traditional flow cytometry in the way that it can provide fluorescently indicated morphological and physiological information in the context of a single cell image [62]. For example, bacterial length and granularity data measured through IFC have been used in a model to predict persistence based on a cell’s morphological features [63,64]. Additionally, through continual improvements to IFC, virtual-freezing fluorescence imaging flow cytometry (VIFFI-FC) has been developed. This technique utilizes a microfluidic chip and timing-based device to allow for a much longer exposure time during imaging, drastically improving image quality. While this technique has only been introduced in eukaryotic cell studies, authors acknowledge its applicability for studying bacterial pathogens [65].

## 3. Microfluidic Devices

Because only some cells from an isogenic population become persisters, it is currently impossible to predict which cells to track in a clonal population. Additionally, considering that persisters are present at very low frequencies in bacterial populations, identifying these cells for further investigation is a challenge. Microfluidic devices, coupled with advancements in cameras and microscope resolution, have been an essential tool to fill this knowledge gap [66]. Over the last fifteen years, these devices have been vastly improved, from their ease of use and affordability to their technical precision to manipulate fluids on a micrometer scale. These advancements have led to novel applications in the field of bacterial persistence, as single cells can be followed through antibiotic treatment and recovery for many generations [66].

This single-cell technology allows exploration of different stresses in order to better understand phenotypic heterogeneity. Persisters can then be found and the data throughout the experimental time course can be used to better predict which cells have the potential to persist. One application of this technique was accomplished by Goormaghtigh and Van Melderen who used a fluorescent reporter to track the genetic expression of the SOS response indicator *sulA* as a measure of DNA damage [67]. There were only 23 persisters in their original population of 47,000 exponentially growing cells, emphasizing the need to study a high quantity of cells before a relevant amount of persister data can be collected.

Antibiotic persistence has been shown to contribute to antibiotic resistance, so evolution is often discussed in the persistence field [6,7,8]. The Mother Machine is a microfluidic device developed by Wang et al. in 2010 to track these changes on a single-cell level [68]. The design consists of multiple channels, each encapsulating a single cell that can obtain nutrients through diffusion of constantly flowing media. The original cell stays secured in the channel as the progeny are forced up, out, and downstream by the flowing media, allowing for study of generations of cells in a high-throughput manner (Figure 3A). This Mother Machine was originally developed for *E. coli*, has since been used for *B. subtilis*, and was recently adapted for cocci morphologies [68,69,70]. Other variations on the Mother Machine have expanded its applicability for exploring the effects of different stresses over generations of growth. One such study showed that biased partitioning of efflux pumps favors the mother cell over the daughter, leading to heterogeneous efflux pump activity and variable antibiotic susceptibility within a clonal population [71]. The dual-input Mother Machine (DIMM) enables generational persistence studies by introducing a secondary, antibiotic-containing liquid in addition to the standard growth media [72]. To manage large amounts of visual data, bacteria Mother Machine analysis (BACMMAN) was developed [73]. This software automates the process of analyzing single-cell images and is currently being expanded to other cell types.

A critical function of microfluidic devices is the precise control of fluid flow through channels. This can be used to execute exact antibiotic gradients and flow rates. Bos et al. applied an antibiotic gradient to single cells in a microfluidic device while tracking filamentation and cell size over time to relate single-cell drug response, morphology, and susceptibility simultaneously [74]. Fluid flow can be manipulated to force cells into specific chambers depending on their size and morphology, thus enabling co-culture experiments without interspecies cross contamination [75]. Original experiments using this approach were performed on larger eukaryotic cells and recent work has adapted the parameters for smaller yeast cells, showing the potential for high-throughput morphological sorting in bacteria [76,77,78]. The flow through microfluidic devices can also be used to direct the development of certain phenotypes. Biofilms can be formed by controlling the laminar flow of planktonic cells around corners [79,80]. Biofilm imaging at a single-cell level has been recently developed, leading to the exciting future potential to combine not only fluorescent probes or reporters, but also antibiotic gradients with microfluidic devices to understand persister physiology within biofilms [81].

Similarly, droplet microfluidics have been used to separate out individual cells but keep them contained in a capsule of liquid. Droplets are formed from the use of two liquids—often water-based media and oil—that do not mix (Figure 3B). By controlling the flow of these two liquids as well as the geometry of their interaction, droplets of different sizes can form. Cells floating inside the media can then become trapped in individual droplet bubbles [82]. If desired, these droplets can be machine sorted via an electrode or with fluorescent reporters to allow for automated quality assurance [83]. Droplets can be combined with each other, injected with new media, and sorted after the cells are inside [84]. Because the liquid of each cell is self-contained, it allows for secretion studies and assays to be performed on single cells [85]. Additionally, work has been carried out showing that washing of cells is possible without disrupting the system, meaning that more complex persistence assays of single cells are on the horizon [86].

Microfluidic devices offer a crucial platform for studying individual cell morphologies, functions, and phenotypes and are amenable to downstream analyses that can uncover more layers of detail within a given cell [87]. Highly technical instrumentation and synthetic biology approaches are helping to bring these molecular-level phenotypic differences to light.

## 4. Mass Spectrometry

Mass spectrometry is a hallmark analytical technique for identifying molecules with high specificity. While studying factors involved in persister formation and resuscitation often involves targeted methods, such as the fluorescent probes discussed previously, mass spectrometry offers a holistic means of studying cellular composition. Mass spectrometry is often utilized for metabolomic and proteomic studies to understand which cellular processes are altered during persister formation and reawakening [88,89,90,91]. The instrumentation and sample preparation protocols for mass spectrometry are diverse, highly technical, and will not be discussed here. Instead, we will focus on the principles of mass spectrometry and how this technique can be applied for inquiries into single-cell persister heterogeneity.

In mass spectrometry, a mixture of molecules is fragmented, ionized, and propelled through an electric field; the time to travel through the field to the detector is then used to calculate the mass of each ion. The fragmentation pattern of the entire sample can then be analyzed to deduce original molecular compositions based on the masses of each atomic element. Because mass spectrometry provides a broad, unbiased snapshot of cell composition, the resultant spectra can be extremely complex. Peak assignment software is continually being improved in order to reduce the burden of data analysis, but it is often a computationally taxing, slow process [92,93,94,95].

To circumvent this issue, a fundamental strategy for studying a specific molecule of interest is stable isotope labelling. Isotopes’ unique mass signatures provide a signal to hone in on amidst dense data sets. Isotopically labeled antibodies can also be used to expedite analysis but, like any antibody-based technique, applicability is limited by the availability of antibodies specific to the molecule of interest. A more direct option is to use isotopically labelled nutrient sources, antibiotics, or other substrates; this has been used with nanoscale secondary-ion mass spectrometry (NanoSIMS) to analyze the metabolic heterogeneity of various single-cell populations [96,97].

Researchers can use high-resolution mass spectrometry imaging techniques to examine the spatial distribution of analytes within single cells. Nanoscale imaging (using cluster ToF-SIMS) has been used to demonstrate the localization of ribosome-targeting versus cell wall-targeting antibiotics in individual *E. coli* cells without the need for substrate labelling [98]. Imaging mass spectrometry can be coupled with imaging fluorescent probes to yield multiple layers of information about a single sample [99]. This is also a strategy to expedite acquisition time: imaging fluorescent markers first allows researchers to focus on areas of interest for subsequent mass analysis, thus increasing the efficiency of a traditionally low-throughput technique.

Interfacing mass spectrometry with other single-cell approaches, such as flow cytometry or microfluidics platforms, is pushing the boundaries of this foundational technique toward new horizons. Beyond isogenic populations, mass spectrometry can offer a highly sensitive platform for studying metabolism in multispecies cohorts. In 2008, Behrens et al. combined fluorescent 16S rDNA probes with stable isotope profiling of carbon and nitrogen substrates to identify which species were responsible for the observed metabolic behaviors in a bacterial cohort [100]. This has inspired a wealth of studies on ecophysiology and offers the potential for studying phenotypic heterogeneity in multispecies contexts [96,101]. But these strategies will only capture intracellular or cell-surface molecular identities; to study single-cell secretomics, droplet mass spectrometry can be employed. As a cell secretes metabolites and enzymes into its surroundings, the analytes will remain associated with that cell due to encapsulation within the same droplet [102]. Therefore, this offers a way to investigate the molecular identities of secreted or excreted products from a single cell and could be relevant to studying the role of antibiotic efflux in persistence [103,104,105].

However, for the foreseeable future, there is one obstacle in mass spectrometry that cannot be avoided: sample destruction. The ionization process renders samples unusable for downstream analysis. Therefore, mass spectrometry is unsuitable for measuring a single cell’s metabolic perturbations over time and tracking its survival through antibiotic treatment and cessation. In eukaryotic single-cell mass spectrometry, microcapillary sampling has been developed as a means to analyze the cytosolic composition of cells without compromising cell integrity; however, as of yet, this technique has not been scaled down for use in bacterial systems [97]. While we await prokaryotic cell microsampling or alternative technological advancements, other non-destructive methodologies are available for metabolic analysis of bacterial persisters.

## 5. Raman Spectroscopy

Raman spectroscopy is an alternative to mass spectrometry for detailed molecular analysis and is a rapidly improving technology for single-cell metabolomics. This technique involves measuring the vibrational bond energies between atoms in a molecule then analyzing the resultant spectrum to glean information on molecular structure. In a seminal 2004 manuscript, Huang et al. demonstrated that Raman spectroscopy could be used for single-cell identification based on the distinct spectral “fingerprint” of each species, including non-culturable environmental isolates [106]. Furthermore, they tracked the spectral peak shifts in cells grown in varying amounts of glucose with heavy carbon (^13^C), highlighting the potential for this technique in studying single-cell metabolic activity. However, efficient implementation of stable isotope-labelled substrates requires the use of chemically defined media that could alter cells’ native metabolic states and limit the applicability of this technique only to bacteria that can grow in laboratory conditions.

An elegant alternative to substrate labelling is to measure metabolism holistically by culturing bacteria in partially deuterated water. This technique, called deuterium isotope profiling by Raman spectroscopy (Raman-DIP), takes advantage of the “silent” region of bacterial Raman spectra: in this range between 2040 and 2300 cm^−1^, there are no measurable intramolecular vibrational energies. Conveniently, carbon-deuterium (C-D) bonds are found in this range. As metabolically active cells incorporate deuterated water molecules into new biosynthetic products, they will create new C-D bonds. These bonds are measurable with minimal signal-to-noise complications and can serve as a global indicator of biosynthetic activity [107]. Raman-DIP is label-free, inexpensive, and fast: cells need only 20 min in deuterated water for Raman spectra to begin showing C-D peaks. Raman-DIP therefore provides a practical approach for studying metabolism, an important driver of the persister phenotype, with minimal experimental perturbations [108].

This approach for metabolic profiling at the single-cell level has proven highly informative for studying microbial phenotypic heterogeneity [109,110,111]. A major wave in the field of persistence comes from the mounting evidence that bacterial persisters are not fully dormant [16,56,112]. Raman-DIP experiments from Ueno et al. supported this hypothesis by revealing that *M. tuberculosis* persisters are non-growing but still metabolically active [113]. Xu et al. used Raman-DIP to elucidate that different intracellular *Salmonella enterica* serovars phenotypically switch from carbohydrate to lipid metabolism for survival within host immune cells and that this switch occurs heterogeneously even within a clonal bacterial population [114]. This is highly relevant to the study of bacterial persisters in the host context because many bacterial species adopt an intracellular lifestyle: *M. tuberculosis* and *Salmonella* species are two classic examples, but the intracellular pathogenicity of other species, such as *S. aureus* persisters residing in macrophages, is still being uncovered [115,116,117,118]. Raman spectroscopy and DIP will continue to play an important role in understanding transient persister phenotypes in a variety of bacterial and host model systems.

Raman spectroscopy is well suited for sorting cells in analytical pipelines because cells remain intact and culturable. For example, Lee et al. created a microfluidics platform for sorting single bacteria based on their Raman spectra, shunting inactive cells to a waste outlet and harvesting metabolically active cells for additional downstream analysis [119]. However, researchers must consider whether transient phenotypic states are perturbed in these multi-step pipelines. Generally, the minimally intrusive methodologies of Raman spectroscopy make it an attractive option for rapid analysis of the metabolic workings within single cells.

## 6. Next-Generation Sequencing

Next-generation sequencing (NGS) allows for untargeted, comprehensive analysis of genetic and transcriptomic information [120,121]. While bulk NGS studies of bacteria have led to discoveries such as the identification of novel species in the environmental microbiota, single-cell NGS applications have the power to shed light on the rare genetic events or low-level transcripts that are overwhelmed by bulk sequencing approaches. Single-cell genomics can also provide valuable insight to the consequences or potential genetic drivers of phenotypic heterogeneity, for example, the downregulation or silencing of DNA mismatch repair genes that increases the mutation rates of affected cells [103,122,123].

In recent years, several research groups have developed strategies to sequence and determine the quantitative levels of RNA transcripts in a single bacterial cell [124,125]. An overarching workflow for these different approaches involves isolating a single bacterium, which can be achieved using some of the techniques described in this review (e.g., FACS and the use of microfluidic devices). Then, segregated cells are lysed for access to their RNA pool, the RNA is reverse transcribed into a cDNA library, and NGS is used to read the transcripts.

Compared to single-cell genomic NGS, single-cell RNA sequencing (scRNA-seq) has been more difficult to optimize in bacteria due to inherent differences between prokaryotic and eukaryotic RNA transcripts. Bacterial RNA transcripts are typically single-stranded, short-lived molecules of extremely low abundance: the average number of a given transcript is estimated at only 0.4 copies per cell [125]. Eukaryotic scRNA-seq methods leverage the poly(A) tail of mRNA transcripts for amplification; however, bacteria lack this RNA processing and require alternative enrichment strategies before sequencing. Without enrichment, the signals from abundant rRNAs and tRNAs will obscure the detection of rare transcripts; therefore, the utilization of exonucleases or Cas9 machinery has been implemented to degrade rRNA or tRNA [126,127]. Blocking primers that recognize and bind canonical rRNA sequences can also be used to prevent further reverse transcription [126,127,128]. These depletion strategies can be complemented by exogenous *E. coli* poly(A) polymerase I that artificially adds poly(A) tails to facilitate subsequent amplification [127,129]. These steps enable relevant messenger transcript enrichment and amplification while decreasing the computational burden of sequence analysis in the end.

For truly untargeted amplification of a bacterial transcriptome, using a set of known primers is inappropriate. One method of total transcriptome amplification is multiple displacement amplification, which utilizes a mix of random hexamers to increase the likelihood of probes hybridizing to every transcript at least once. After reverse transcription, additional random hexamer primers are added, directing the phage DNA polymerase Φ29 to elongate the complementary strands over several amplification cycles [128,130]. While transcript amplification using Φ29 has only been applied to transcriptomic analysis via microarray, single primer isothermal amplification (SPIA) is a method that yields ample, clean cDNA suitable for scRNA-seq (Figure 4A) [131]. However, these methods can introduce amplification bias because cDNA of more abundant transcripts becomes exponentially more prevalent with each round of amplification.

To combat this issue, Sheng et al. developed multiple annealing and dC-tailing-based quantitative scRNA-seq (MATQ-seq) for amplifying all RNA transcripts while also reducing amplification bias (Figure 4B) [132]. The principle underlying MATQ-seq’s increased transcriptome coverage is the use of common probes that, at low temperatures, can hybridize anywhere along RNA transcripts to initiate reverse transcription. This technique was recently applied for determining how bacterial single-cell transcriptomes vary depending on growth state [133]. MATQ-seq is also quantitative, meaning that it allows comparison of transcript levels between cells, not just comparison of the relative transcript levels within a given cell. To circumvent cell-to-cell amplification variance during analysis, MATQ-seq utilizes an amplicon normalization strategy that divides each transcript’s abundance by the cell’s total amplified RNA [132]. MATQ-seq also leverages unique molecular identifiers (UMIs) to aid in quantification; UMIs are random hexamers ligated onto each cDNA template before amplification [132,134]. During data analysis, the prevalence of certain UMIs over others can be used to elucidate the effects of amplification bias versus actual transcript abundance variations.

Many scRNA-seq protocols begin with single-cell isolation and lysis but, because some bacterial species are encapsulated by a rigid cell wall, lysing single bacteria presents an expensive and labor-intensive challenge [135]. To circumvent the technical hurdle of lysing individual cells with miniscule proportions of reagents, split-pool barcoding can be implemented instead (Figure 4C) [129,136]. This technique involves labeling each cell’s transcripts with a three-part barcode, resulting in nearly one million possible barcode combinations [129,136]. Lysis, amplification, and sequencing can then be performed on all cells en masse because each read will have a barcode ascribing it to its cell of origin [137]. Split-pool barcoding is featured in recent scRNA-seq protocols such as microSPLiT and PETRI-seq that hold promise for studying transcriptional heterogeneity in bacterial populations [129,136]. For example, Blattman and colleagues demonstrated the power of PETRI-seq by detecting an instance of rare gene induction that occurred in only 0.4% of cells in a population of *S. aureus* [136]. However, split-pool barcoding should be limited to analyzing roughly 10,000–30,000 cells or the risk of repeating barcode combinations in multiple cells increases, possibly confounding single-cell identification. Considering the rarity of persisters under certain growth conditions, additional barcoding steps may be required to increase the unique combinations and the number of cells that can be analyzed. Overall, this technique makes single-cell sequencing more accessible for laboratories without FACS capabilities, single-cell manipulators, or microfluidic devices while increasing throughput compared to many single-cell isolation protocols.

The advancements discussed above have allowed for large-scale analyses of the phenotypic states and genetic determinants underlying bacterial persistence, but further optimization is needed for bacterial scRNA-seq to be as accessible and reliable as scRNA-seq in eukaryotes. Continued improvements in bacterial isolation and lysis, mRNA enrichment, library amplification, and sequencing protocols can broaden transcriptome coverage, improve assignment to single bacteria, and ease experimental and/or computational workflows.

## 7. The Future of Studying Single-Cell Histories

Innovative advances in biological engineering have given new life to familiar fluorescence-based techniques for exploring persister physiology. Beyond understanding the status of a single cell at a moment in time, the cutting-edge technologies highlighted here can report on generations of cell division without the need for direct, time-lapse observation. The difference in division rates of clonal bacterial cultures is fundamental to persister formation and resuscitation. Previously, Roostalu et al. investigated the rate of bacterial division and its role in persistence by inducing a parent population to express GFP and then measuring how the GFP signal decreased with successive generations [138]. In the span of only two hours, the GFP signal of individual exponential-phase *E. coli* was nearly diluted to uninduced/baseline levels, demonstrating the limits of this approach to measuring cell division over a longer time span.

In order to measure generations of cell division in individual bacteria, synthetic biologists have designed various intracellular “clocks”, the newest development coming from Riglar et al. with the Repressilator 2.0 (Figure 5A) [139,140,141]. The improved Repressilator circuit reliably controls fluorescence expression in a cycle that is independent of cell growth or time. The cycle fluctuates based on cellular divisions, allowing researchers to determine the number of bacterial generations occurring between fluorescence measurements without the need for continuous sampling or observation (Figure 5B). Traditional use of fluorescent reporters is limited to reporting on the current state of the cell; the beauty of Repressilator-like technology is the ability to see the growth history of a single cell for longitudinal or in vivo studies of persistence. Riglar et al. used this tool in antibiotic-treated mice to show that pathogenic bacteria divide rapidly upon introduction to a barren gut and that, as the gut is recolonized, fewer generations occur between sampling points [139]. Beyond reliable reporting of bacterial divisions, oscillatory circuits such as the Repressilator could be used for phase-tuned gene expression to study how the timing or fluctuation of gene expression affects bacterial persistence in vivo.

Another notable development in cellular recording comes from Farzadfard et al. with DOMINO: the DNA-based Ordered Memory and Iteration Network Operator [142]. This system utilizes gene-editing enzymes that edit specific nucleosides on the chromosome to, essentially, use base pair conversion as the binary 1’s and 0’s of computer code (Figure 6A). The enzymes are directed to edit specific sites by guide RNAs (gRNAs) that are under the control of inducible promoters. When a stimulus induces gRNA expression and gene editing, the resultant base pair alterations become part of the recorded cellular “memory”. The altered base pairs can, in turn, trigger additional effects such as fluorescent protein expression so that cell memories can be “read” without requiring destructive sequencing (Figure 6B). The system can also be programmed with various logic frameworks for recording the synchronicity of multiple stimuli, the temporality of step-wise exposures, and more (Figure 6C). The ability to sort cells by FACS based on their histories (recorded genetically, reported fluorescently) and then resume culturing until later memory interrogation enables longitudinal monitoring of single-cell exposures and their impact on phenotypic heterogeneity. Furthermore, programming logic circuits to not only record cell memory, but to actually control gene expression, allows for fine-tuned experimental interrogation. However, because this system relies on a limited arsenal of tightly controlled inducible promoters, the ability to study a variety of signals and a broader range of signal induction intensities is currently lacking. We anticipate that advancements in rational promoter design or transcriptional regulators such as riboswitches would make this system more applicable to studying biologically relevant pathways, such as DNA damage repair and intercellular signaling, that are implicated in antibiotic persistence [143,144,145,146]. Further development of synthetic biology tools for use in memory-recording systems such as DOMINO could revolutionize how we investigate single-cell physiology entirely.

## 8. Clinical Applications of Single-Cell Techniques

While the incorporation of many of these cutting-edge, single-cell techniques has revolutionized the study of bacterial physiology, the techniques are also expanding the diagnostic capabilities of clinical medicine. Traditional, population-based assays used in clinical microbiology labs, such as minimal inhibitory concentration (MIC) assays to identify antibiotic resistant strains, do not allow for the detection of tolerant or persistent organisms that contribute to relapsing infections [2]. While the classification of persistent bacterial populations in a clinical setting is not yet common practice, many groups have begun to incorporate single-cell techniques and NGS to unveil heterogeneous antibiotic sensitivities and improve patient treatment regimens.

The incorporation of microfluidic devices in parallel with imaging provides new opportunities for pathogen identification and antibiotic susceptibility testing at the single-cell level, significantly decreasing the time from sample collection to diagnosis. After collecting a septic patient’s blood sample, blood cells are removed by centrifugation and the supernatant—the bacteria-containing fraction—can be concentrated and loaded into a microfluidic device to isolate, visualize, and test individual bacteria [147]. A microfluidic device with adjustable channel heights can classify bacterial pathogens in a sample by their morphologies [148]. Embedding oxygen-sensing nanoprobes into the design allows additional reporting on metabolic activity [149]. Subsequent antibiotic susceptibility testing (AST) can be accomplished with continuous imaging tracking the growth and reproduction of single cells in increasing concentrations of antibiotics. The methods of swift identification and phenotype testing are exceptionally useful in complicated cases, such as sepsis. Bacterial cultures from septic patients can take anywhere from 5 to 7 days to analyze, costing precious time in which the patient’s condition can rapidly deteriorate. Microfluidic apparatuses enable detection of antibiotic resistance in as little as 3 h [147].

In addition to the applicability in sepsis models, microfluidic devices can utilize a fluid droplet system to identify slow-growing bacteria with greater sensitivity than traditional culture techniques. Fastidious anaerobic pathogens of the gut, such as *Clostridioides difficile*, can be exceptionally problematic and impervious to antibiotic therapies. Droplet-based technologies offer more sensitive detection because single microbes are aliquoted into liquid droplets where their growth and division can be assessed over time and in various media conditions [150]. These technologies function as a tool for the identification of pathological bacteria by systematically assessing bacterial size, growth rate, antibiotic susceptibility, or genetic material within given patient microbiomes and/or disease states. Interrogating genetic sequences and morphological features of bacteria following microfluidic enrichment can provide additional information for understanding a microbe’s pathological potential.

While microfluidic devices are powerful on their own, their amalgamation with cutting-edge spectroscopic or NGS techniques enhances their analytical power. Liu and colleagues developed a silver nanorod substrate serving as a tool to establish pathogen chemical fingerprints by surface enhanced Raman spectroscopy [151]. This tool enables the identification of known pathogens in a complicated microbiological milieu. Additionally, Raman spectroscopy can be used to concomitantly assess pathogen identity and metabolic activity, which can offer insight into a given cell’s antibiotic sensitivity. Fast Raman-assisted antibiotic susceptibility testing detects deuterium incorporation by bacteria in the presence of antibiotics, allowing clinicians to infer susceptibility based on the measured metabolic rates [152].

Antimicrobial sensitivity and metabolic activity of individual bacterial cells can also be deduced using NGS. DropDx is a microfluidic-based technique in which single bacteria are encapsulated in droplets and briefly exposed to antibiotics before thermal lysis [153]. Then, these droplets are incubated with fluorescent probes designed to hybridize to genes encoding 16S rRNA of bacterial pathogens. This method operates under the assumption that 16S rRNA will be more abundant in droplets containing growing populations than those with non-growing groups; therefore, resistant pathogens growing in the presence of antibiotics will have higher abundance of 16S rRNA and higher fluorescent readout [153]. As antibiotic refractory bacterial infections are becoming increasingly urgent, it is even more essential to obtain fast and accurate diagnoses. Grumaz and colleagues found that, compared to traditional culturing methods, NGS could consistently detect bacteria circulating in patient blood at a six-fold higher positivity rate throughout the course of clinical management [154]. The advent of more efficient, cost-effective, and accurate single-cell technologies for implementation in hospital settings will enhance clinical decision making on optimal antibiotic regimens and the best practices to improve patient outcomes [155,156].

## 9. Conclusions

Single-cell technologies have shown immense utility in studying antibiotic persistence phenotypes as well as other manifestations of phenotypic heterogeneity. Fluorescence-based techniques utilized in tandem with flow cytometry, FACS, and microscopy are foundational tools for measuring the relative abundances of nucleic acids and proteins, the localization of biomolecules, cellular morphology, and signaling transduction. Mass spectrometry and Raman spectroscopy have further increased the resolution of metabolomics and proteomic investigations. Advancements in microbial NGS technologies, specifically scRNA-seq, have been vital in the holistic identification and analysis of expressional trends at the single-cell level. Finally, microfluidic devices provide high-throughput single-cell platforms for studying growth dynamics, division, metabolism, and more. Interfacing these techniques with one another strengthens our ability to bridge knowledge gaps of persister physiology.

Even though these highlighted techniques have significantly contributed to studying phenotypic heterogeneity, there are still shortcomings that could be addressed. To capture morphological heterogeneity across a bacterial population, we need high-resolution imaging at higher throughput. VIFFI-FC is an emerging technique which intends to solve this problem; however, VIFFI-FC has only been used to study eukaryotic systems and requires testing and validation in prokaryotes [65]. Beyond monocultures, persisters in multi-species communities and biofilms remain challenging to characterize. Repurposing of single-cell techniques (such as the development of par-seqFISH to study spatial transcriptomics) can accelerate research on the persisters of complex microbial communities [36]. Additionally, many techniques are limited to measuring a cell’s present state and may require cell destruction for analysis. We can employ systems such as the Repressilator 2.0 and DOMINO to shed light on single-cell growth histories and the timing of expressional events through longitudinal and/or in vivo experiments [139,142]. The refinement of these techniques for use in various bacterial species and with various native promoters will further enhance our capacity to study heterogeneous phenotypes and single-cell physiology.

We anticipate that the insights gained into bacterial phenotypic heterogeneity using these emergent single-cell techniques will also prove informative to relevant eukaryotic systems such as cancer. Many of the same principles governing bacterial persistence can be applied to parallel investigations into cancer persister cells; on the other hand, breakthroughs in eukaryotic single-cell technologies can accelerate the development of finer-resolution tools for prokaryotes [157,158,159]. There is a clear mutual benefit to advancing these seemingly distinct fields of research. Each advancement in single-cell technology opens new avenues of investigation into persister physiology, helping us realize the broader impact of phenotypic heterogeneity in prokaryotic and eukaryotic systems alike.

## Figures and Tables

**Figure 1 microorganisms-09-02277-f001:**
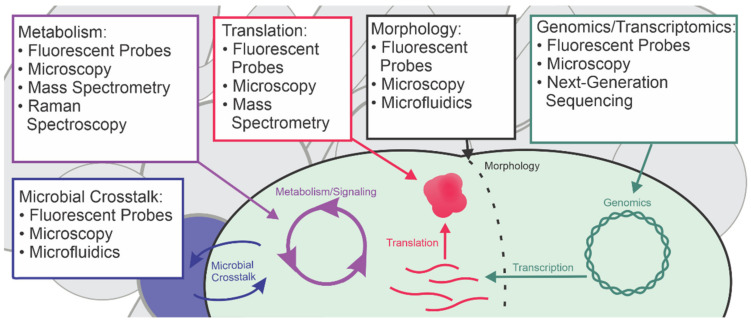
An overview of approaches to investigate single-cell physiology.

**Figure 2 microorganisms-09-02277-f002:**
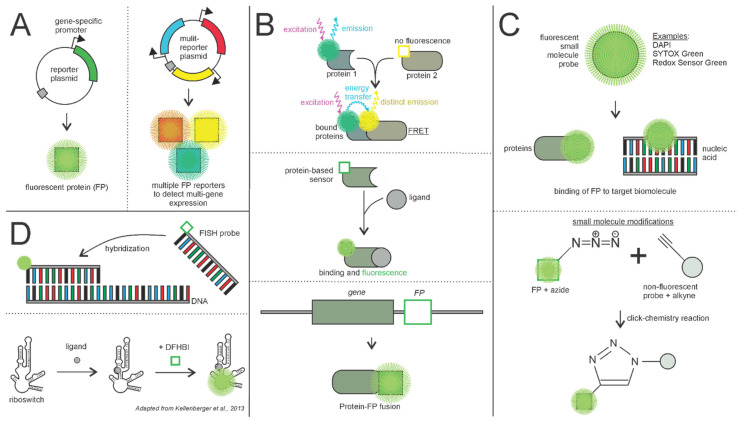
Principles of fluorescence-based techniques. (**A**) Plasmid-based fluorescent reporters utilize gene-specific promoters tied to FP molecules to indicate expression. This concept has been further refined to make multi-reporter plasmids, where the expression of multiple genes can be observed simultaneously. (**B**) FRET-based probes can be incorporated into biomolecules to indicate when the two interact. Other protein-based and enzyme-based sensors can fluoresce upon binding of a ligand (for example, ATP biosensors such as QUEEN and iATPSnFR). FP tags can be incorporated into a chromosomal or plasmid-borne gene copy to create a fluorescent protein-FP fusion. (**C**) Fluorescent small molecule probes such as DAPI, SYTOX Green, and Redox Sensor Green can directly bind target biomolecules and indicate their presence and localization. Non-fluorescent molecular probes can be modified by alkynation so that they fluoresce upon reacting with an azide-containing fluorescent protein. (**D**) Nucleic acid-based fluorescence techniques, such as FISH probes, can bind to complementary nucleic acid sequences, which fluoresce upon hybridization of the two strands. Riboswitches, such as Spinach, can change conformation upon binding to a ligand to allow incorporation of a fluorescent molecule such as DFHBI.

**Figure 3 microorganisms-09-02277-f003:**
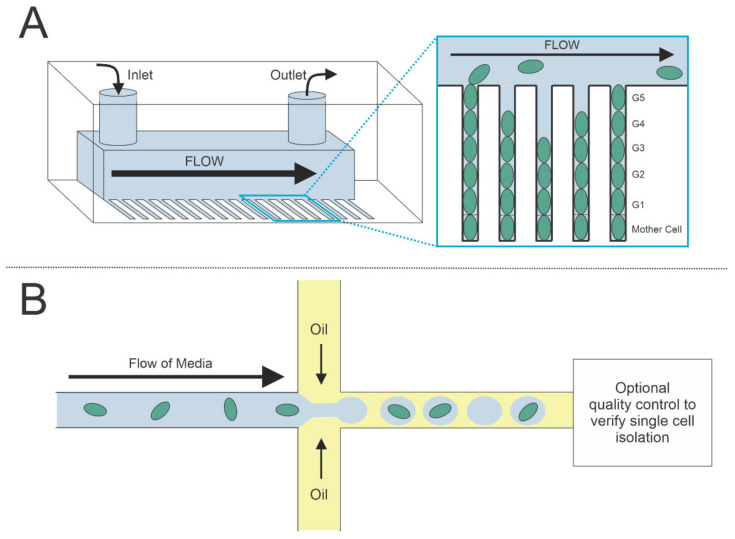
Highlighted applications of microfluidic devices. (**A**) The Mother Machine is a microfluidic device housed in a chip that allows media to flow through an inlet site and exit via the outlet [69]. Cells are inoculated in channels that are designed to ensure the entrapment of the oldest cell (“Mother Cell”) of the lineage. These initial cells give rise to progeny over time which can be studied. The media flows constantly to ensure the cells are fed and will continue to replicate while also allowing for the removal of old progeny that outgrow the rows. (**B**) Droplets containing single cells can be formed by controlled flow of oil around cells in media. The oil and water-based media do not mix, so the cells stay in their encased bubbles. After the initial droplet formation, droplets can be further analyzed fluorescently or electrically to ensure only a single cell is present.

**Figure 4 microorganisms-09-02277-f004:**
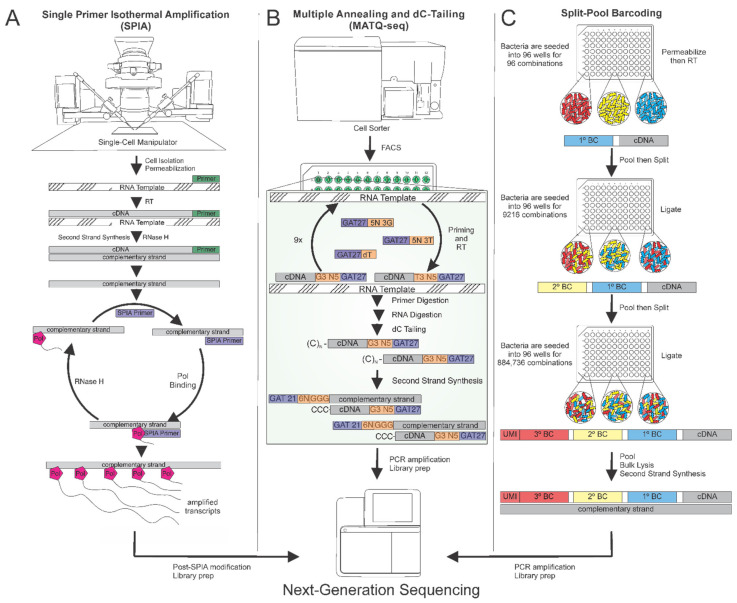
Approaches to Next-Generation Sequencing library preparation. (**A**) After single-cell isolation (by a single-cell manipulator, for example), cell lysis, reverse transcription, and first strand synthesis, Single Primer Isothermal Amplification (SPIA) is conducted using the SPIA primer and polymerase for linear amplification. Then, bacterial transcripts are modified, purified, and ready for library preparation. (**B**) MATQ-seq requires isolation of a single bacterium (by FACS, for example) and cell lysis. RNA templates are reverse transcribed into cDNA using primers that primarily contain G, A, and T bases (GAT27 primers). These complementary strands are given dC-tails by TdT terminal transferase. Finally, second strand synthesis is accomplished with primers that recognize and extend from the poly(C) tail. (**C**) Split-Pool Barcoding utilizes cellular barcodes to match transcript sequences to individual cells. Cells are permeabilized in batch culture then seeded into a 96-well plate with a unique primer set in each well. Reverse transcription of RNA templates with these primers results in cDNA strands with a primary barcode attached. Bacteria are then pooled and redistributed into a different plate twice more for secondary and tertiary barcode addition. Each tertiary barcode includes a UMI, a randomly generated hexamer which correlates to a single cDNA transcript. Then, bacteria are pooled for bulk cell lysis, transcript amplification, and library preparation.

**Figure 5 microorganisms-09-02277-f005:**
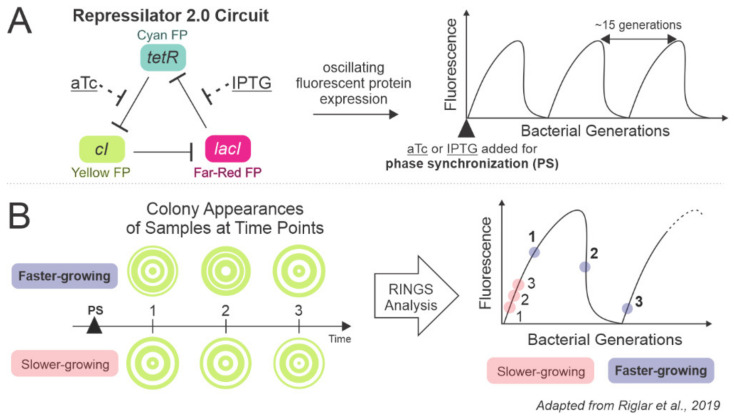
Repressilator gene expression oscillates with cell divisions. (**A**) The Repressilator 2.0 circuit involves three genes and associated fluorescent proteins (FP) in an inhibitory feedback loop. Each gene’s expression rises and falls once approximately every 15 bacterial divisions, independent of time or cell growth rate. Once repression of *cI* is relieved, for example, yellow fluorescent protein also begins to be expressed as a measurable indicator of Repressilator phase. Isopropyl β-D-1-thiogalactopyranoside (IPTG) or anhydrotetracycline (aTc) are used to phase synchronize (PS) a population to the same phase of the circuit. (**B**) Single cells are sampled from a population by plating. As the cell divides and expands into a colony, the outward growth forms a pattern of fluorescent rings. Riglar et al. developed the Repressilator Inference of Growth at the Single-cell level (RINGS) workflow for analyzing colony images and attributing each fluorescence pattern to a generational phase [139]. Bacterial growth rate can be inferred by the phase changes between time points.

**Figure 6 microorganisms-09-02277-f006:**
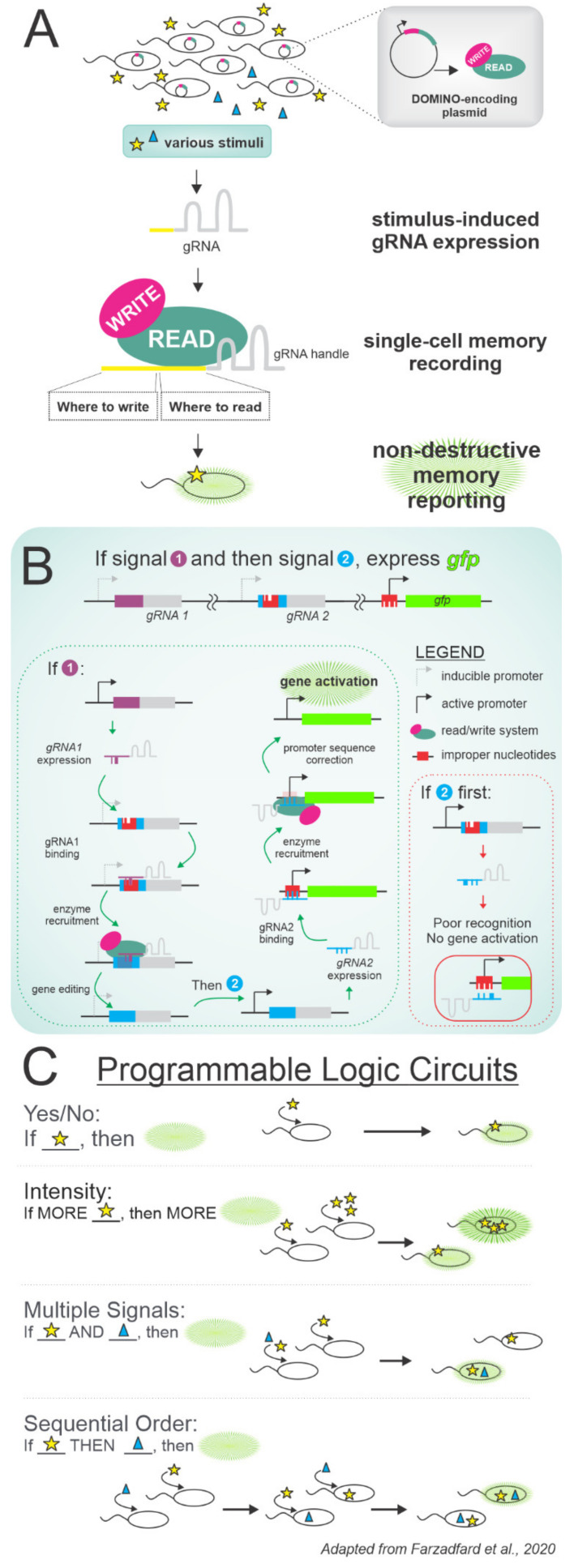
DOMINO is a tool for single-cell memory recording and reporting. (**A**) Stimulus-induced gRNAs direct the cytidine deaminase- and Cas9-based read/write system to edit specific base pairs in the chromosome. These edits can turn on the expression of fluorescent proteins for non-destructive memory reporting. (**B**) DOMINO circuitry can be programmed using gRNAs that target specific sites on the chromosome for gene editing. Chaining events of gene expression and editing together builds complexity beyond simple recording or reporting. This panel’s schematic details an example of sequential logic. (**C**) Logic circuits that have been demonstrated with DOMINO include reporting on stimulus reception, stimulus intensity, multiple stimuli, and sequential reception.

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
