# Peer review of "Single-Cell Technologies to Study Phenotypic Heterogeneity and Bacterial Persisters"

_microorganisms, 2021, doi:10.3390/microorganisms9112277_

Round 1
Reviewer 1 Report
In this review, the authors have summarized the recent developments and technologies developed towards studying bacterial persistence. The authors cover all the techniques from flow cytometry, mass spectrometry, microfluidics to recent developments in single-cell sequencing. The published literature are well cited and context provided. The illustrations are also very well thought of to describe the important concepts. I am quite satisfied with the overall content and breadth of the review in its current form.
Author Response
We would like to thank Reviewer 1 for their review of our manuscript and positive feedback.
Reviewer 2 Report
The review manuscript entitled "Single-cell technologies to study bacterial persister heterogeneity" does present a review about heterogeneity, not persister heterogeneity. The title should be corrected. The majority of the methods presented here make alter cellular status of cells, which can not be applied for persisters which have metastable phenotypes.
Besides, the quality of the figures should be improved, when the manuscript was printed, the figures cannot be read.
It would be better to organize figure 1 so that the upcoming sections of methods can be also followed using the figure 1.
Author Response
We would like to thank Reviewer 2 for their review of our manuscript. We have modified our review according to the reviewer’s suggestions. Our point-by-point responses are presented below:
- We have modified our title to “Single-Cell Technologies to Study Phenotypic Heterogeneity and Bacterial Persisters” to account for the reviewer’s suggestion that the techniques are applicable for investigating cellular heterogeneity in general.
At the same time, we would like to note that the techniques highlighted in this review were chosen with a focus on studying persistence as one facet of phenotypic heterogeneity. We present examples where single-cell techniques were previously used to investigate persistence, including Raman-DIP experiments and the use of fluorescent reporters:
Lines 406-410:
A major wave in the field of persistence comes from the mounting evidence that bacterial persisters are not fully dormant [16,57,113]. Raman-DIP experiments from Ueno et al. supported this hypothesis by revealing that M. tuberculosis persisters are non-growing but still metabolically active [114].
Lines 535-538:
The difference in division rates of clonal bacterial cultures is fundamental to persister formation and resuscitation. Previously, Roostalu et al. investigated the rate of bacterial division and its role in persistence by inducing a parent population to express GFP and then measuring how the GFP signal decreased with successive generations [139].
We also suggest that many techniques for studying single cell physiology can potentially be applied in future studies of persistence, citing specific examples that relate to outstanding questions regarding bacterial persistence:
Lines 164-167:
Kellenberger et al. leveraged riboswitch biology to engineer a probe for detecting intracellular cyclic di-GMP, a signaling molecule with critical roles in regulating virulence, planktonic versus biofilm lifestyles, and antibiotic persistence [48,49].
Lines 309-311:
Additionally, work has been done showing that washing of cells is possible without disrupting the system, meaning that more complex persistence assays of single cells are on the horizon [87].
Lines 365-367:
Therefore, [droplet mass spectrometry] offers a way to investigate the molecular identities of secreted or excreted products from a single cell and could be relevant to studying the role of antibiotic efflux in persistence [104–106].
While we agree that many approaches could be further improved to study transient phenotypes, we believe that our examples highlight their utility in studying heterogeneous phenotypes in bacteria, including those pertaining to bacterial persistence.
- We acknowledge that the techniques presented here have their limitations, including perturbations to the status of cells in a metastable state as the reviewer has pointed out. In our introduction, we have expanded on this concept and state that “phenotypic states are inherently transient and shift in response to environmental conditions; therefore, it is even more important that chosen techniques faithfully capture physiological states with minimal cellular perturbations [10]” (lines 50-53). When we discuss specific techniques, we also address this limitation where applicable. For instance, in lines 98-100, we comment that the use of protein fusions can potentially perturb protein folding and localization. In our discussion of the use of Raman spectroscopy (lines 425-426), we stated that “researchers must consider whether transient phenotypic states are perturbed in these multi-step pipelines.” Our sense is that these comments help address the limitations of existing technologies.
- We have exported our figures as >300 dpi TIFF files and we have increased the text size in our figures where needed, which should improve their quality and resolution in the final manuscript.
- We have reorganized Figure 1 such that the techniques that are presented in each box are listed in the order of their appearance in the main text as Reviewer 2 recommends.
Reviewer 3 Report
In the manuscript ID: microorganisms-1418641 by Hare et al., the authors present a review of the last-developed and cutting-edge technologies involved in single-cell studies, with the particular focus on the description of the bacterial subpopulations known as persisters. In reviewing each technique, the authors report the basic principles, thus not resulting redundant with other, more specific reviews, highlight both strengths and weaknesses and underline the specific application in the study of antibiotic persistence. Finally, indications of possible applications of these technologies in the clinical settings, as well as future perspectives are proposed.
The paper is well written and clear; it analyses in a focused perspective the revised techniques, thus offering a wide and specific overview of the latest improvements in single cell and persistence studies. Considering the still limited knowledge about the topic and the related analysis techniques, the review offers a valuable contribution and fits with the aims of the special issue “Novel Approaches for Investigating Antibiotic Resistance and Bacterial Persistence”. The manuscript can be considered eligible for publication in “Microorganisms”.
MINOR COMMENTS
Line 32, please correct “genetically susceptible”;
Line 50, please correct “0.001”;
Line 234, please correct “pass into persisters” or “become persisters”;
Lines 543, 548 and 700, please type “in vivo” in italic.
Author Response
We would like to thank Reviewer 3 for their thorough review of our manuscript and positive comments. We have addressed all of the reviewer’s suggested edits as indicated below, which we think have improved the quality of our review.
MINOR COMMENTS
Line 32, please correct “genetically susceptible”;
We have modified this sentence according to the suggestion.
Line 50, please correct “0.001”;
We have added a zero in front of the number.
Line 234, please correct “pass into persisters” or “become persisters”;
We have changed “transition into persisters” to “become persisters” as recommended.
Lines 543, 548 and 700, please type “in vivo” in italic.
We have italicized “in vivo” in these sentences.